# Internet-based support for informal caregivers to individuals with head and neck cancer (Carer eSupport): a study protocol for the development and feasibility testing of a complex online intervention

Ulrica Langegård [1], Åsa Cajander [2], Maria Carlsson,[3] Louise von Essen,[4] Awais Ahmad,[2] Göran Laurell [5], Ylva Tiblom Ehrsson [5], Birgitta Johansson[1]

YTE and BJ contributed equally.

For numbered affiliations see end of article.

**Correspondence to**
Ulrica Langegård;
ulrica.langegard@igp.uu.se

## ABSTRACT

**Introduction** It is strongly recommended that randomised controlled trials are preceded with an exploration of the needs of the target population and feasibility testing of the intervention. The present study protocol is set out to describe these steps in the development of a complex intervention.

The past decades' transition of care from inpatient to outpatient settings has increased the complexity of caregivers' responsibilities, which they may not be prepared for. There is a need to support informal caregivers (ICs) to prepare them for caregiving and decrease the caregiver burden. The main aim of this study is to describe the development of an internet-based intervention (Carer eSupport) to improve ICs' ability to support individuals with head and neck cancer and to describe the testing of the feasibility and acceptability of Carer eSupport.

**Methods and analysis** This is a multicentre study involving the ear, nose and throat clinics and the oncology and radiotherapy clinics at three university hospitals. The study protocol comprises two phases, *development* and *feasibility testing*, using the Medical Research Council framework for developing a complex intervention. Carer eSupport will be based on the results from focus group discussions with ICs and healthcare professionals (planned for n=6–8 in respective groups) and scientific evidence, the Social Cognitive Theory and the Theory of Acceptance and Use of Technology. The feasibility testing will include 30 ICs who will have access to Carer eSupport for 1 month. The feasibility testing will be evaluated with a mixed-method design.

**Ethics and dissemination** All procedures have been approved by the Ethics Committee at Uppsala University (Dnr: 2020-04650). Informed consent will be obtained before enrolment of patients, their ICs and healthcare staff. The feasibility testing is registered at Clinicaltrials.gov (Identifier: NCT05028452). Findings will be disseminated in peer-reviewed journal publications.

**Trial registration number** Clinicaltrials.gov (Identifier: NCT05028452).

## STRENGTHS AND LIMITATIONS OF THIS STUDY

⇒ Carer eSupport will be developed in collaboration with informal caregivers and healthcare professionals, and based on scientific evidence and established theories.

⇒ The feasibility and acceptability of Carer eSupport will be tested using a mixed-method design.

⇒ A thorough systematic review has not been conducted, this may result in loss of valuable research data in comparison to a systematic review.

⇒ Carer eSupport will be provided via a research web portal, developed using a user-centred design.

## INTRODUCTION

It is strongly recommended that a randomised controlled trial (RCT) is preceded by an exploration of the needs of the target population and feasibility testing of the intervention.[1] The present study protocol is set out to describe these steps in the development of a complex intervention that subsequently will be evaluated in an RCT.

## BACKGROUND

Informal caregivers (ICs) are crucial in the care and support of patients with cancer. An IC may be a relative, friend or partner who provides (physical/emotional/practical) assistance to a patient.[2] The shift from inpatient to outpatient cancer care has resulted in a transition of care to ICs, increasing the complexity of their responsibilities, which they may not be prepared for.[3] Thus, ICs may need support.

This study protocol describes the design of a project, Carer eSupport, focusing on

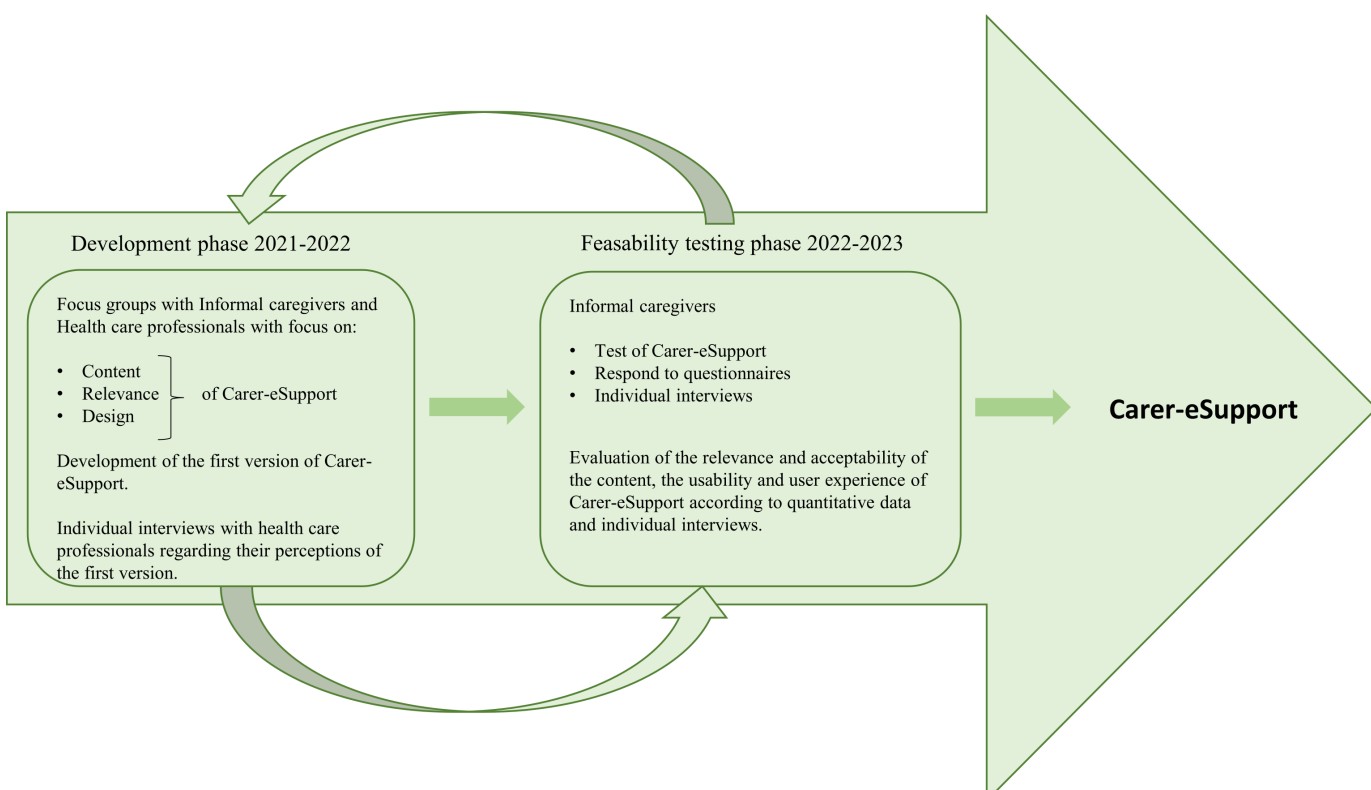

**Figure 1** Overview of the development phase and the feasibility testing phase of the intervention, Carer eSupport, according to Medical research councils guidelines for complex intervention.

ICs to individuals with head and neck cancer (HNC). Carer eSupport is designed using the Medical Research Council framework,[4] which includes four critical phases: (1) development, (2) feasibility testing, (3) evaluation and (4) implementation. This paper describes the first two phases for Carer eSupport (figure 1). A forthcoming RCT will be conducted subsequently to evaluate the effects of Carer eSupport. To ensure a future implementation of the intervention, provided that Carer eSupport is effective, we will conduct a process evaluation.[5] This will make it possible to identify internal (eg, intervention design) and external factors (eg, stakeholders, available resources) that may affect outcomes, and the implementation process.[6 7] Tentative models to be used in the process evaluation (planning in progress) is the logic model, Precede–Proceed planning model[8] and the Information Systems Success Model,[9] to cover IT-specific questions.

## Patients with HNC and their ICs

Over 800 000 people worldwide were diagnosed with HNC in 2018.[10] Many individuals with HNC experience symptom burden due to both the cancer and the treatments they receive, for example, eating problems, pain, difficulty swallowing and speech problems[11 12] and physical and/or psychological distress.[13] This affects not only the patients, but also their ICs. Unmet needs have been identified among 60%–70% of patients and a similar proportion of their ICs.[14] The ICs' responsibilities of symptoms management, including dysphagia management, preparation of

special meals and tube feeding, are time-consuming and may affect ICs' health negatively.[15 16] Therefore, ICs may require support related to challenges along the trajectory, including skills training and psychological and social needs.[17] A recent study concerning ICs of individuals with HNC revealed that the perceived difficulty of caregiving tasks was a more crucial driver of psychological distress than the number of tasks or patient experiences of symptom clusters.[18] Lee *et al*[19] observed that depressive disorder was the most prevalent psychiatric diagnosis in ICs to patients with HNC. Taken together, this indicates a need to develop support interventions for ICs of individuals with HNC.

### A rationale for Carer eSupport

#### Preparedness and caregiver burden
Preparedness for caregiving can be defined as ICs' perceived capacity to provide physical, emotional or practical care and manage the stresses of caregiving.[20] Lacking preparedness may increase caregiver burden, which can be described as a multidimensional biopsychosocial reaction resulting from an imbalance between care demands and a caregiver's care resources.[21] The caregiver burden may encompass physical, mental[22 23] and financial components,[24] and social isolation.[25] High perceived preparedness is associated with a lower caregiver burden and may prevent declining IC health[26] and increase patient care capacity.

## Caregiver interventions

The effectiveness of interventions focusing on the ability to care for patients with cancer has been evaluated in systematic reviews.[27–29] Northouse et al[27] found that interventions for ICs' increased preparedness for caregiving, reduced caregiver burden and anxiety and improved family relationships. Applebaum and Breitbart[28] concluded that interventions combining different types of support, for example, skill-building and psycho-education, including support to cope with the illness, seemed to offer the greatest benefits. Ferrell and Wittenberg[29] showed that pain management skills are particularly important for caregivers, and suggested a focus on interventions focusing on general physical care and symptom assessment. A combination of interventions including instruction assessing side effects, techniques for assisting in pain management and general training in problem-solving skills may increase ICs' preparedness for caregiving.[29] Aung et al[30] suggested an educational programme to prepare caregivers for the complexity of their role. D'souza et al[31] showed that partners of patients with HNC, who followed a tailored education programme, reported higher satisfaction with information provision and less anxiety and depression than those who did not follow the programme. However, while various skill-based interventions have been evaluated, it remains unclear how interventions for ICs of patients with HNC may be designed to provide optimal support.

## Internet-based interventions

Internet-based healthcare interventions can act as a complement to standard care,[32] providing increased availability of information, education and support, especially for people with limited possibilities to visit a clinic. In addition, internet-based interventions may be accessed anytime and when most needed. Thus, internet-based interventions for ICs need to be further studied.[33] There is a large variation in perceived meaningfulness of internet-based activities, highlighting the importance of adapting interventions to the needs and preferences of the intended end users. Willingness to use internet-based interventions depends on early and thorough involvement of end users in design and development.[34–36] Ugalde et al[37] discussed the lack of actions to facilitate implementation in clinical settings. Crucial characteristics of a successful intervention are its relevance for the target groups and its potential to be effectively implemented in clinical care, provided that it is effective. Craig et al[1] highlighted that early and continuous involvement of stakeholders in the development and feasibility testing of an intervention is necessary to achieve these goals. Thus, ICs will be involved in the entire process of the development and testing of Carer eSupport. In addition, healthcare professionals' (HCPs) perceptions of the intervention will be explored to facilitate an eventual implementation in clinical care after the forthcoming RCT has been conducted. Further, artificial intelligence (AI) may create new opportunities for personalised and predictive healthcare systems.[38 39]

Carer eSupport can potentially be integrated with AI to provide ICs with relevant, accurate information, using users' experiences to increase ICs' preparedness for caregiving and decrease caregiver burden.

## Theoretical framework

The Social Cognitive Theory (SCT)[40] is used as a theoretical framework for the development of Carer eSupport. The foundation for SCT is the significance of the interplay between behaviours and personal/social/environmental factors.[40] Self-efficacy is a core construct in SCT, conceptualised as a person's self-perceived capacity to perform in a particular situation.[41] In the context of caring for patients with HNC, it can be assumed that ICs with high self-efficacy will perceive themselves as prepared to perform caregiving tasks, and therefore will be successful in doing so. SCT is supplemented with the well-established Unified Theory of Acceptance and Use of Technology (UTAUT) framework.[42] Technology acceptance models have been designed to predict behaviours and estimate acceptance and satisfaction in individuals using technology. The UTAUT model can also increase understanding of non-use of technology. UTAUT includes four relevant constructs: performance expectancy, effort expectancy, social influence and facilitating conditions. In this study, the constructs are given the following meanings:

1. Performance expectancy refers to whether ICs believe that Carer eSupport can improve their preparedness for care.
2. Effort expectancy refers to the efforts ICs expect when using Carer eSupport.
3. Social influence refers to whether important people in an IC's surroundings believe the IC should use the intervention.
4. Facilitating conditions refers to enablers and barriers that may influence an IC's understanding of the ease or challenges of using Carer eSupport. These could be existing organisational or technical infrastructures that support use of the system.

In summary, the needs of ICs in providing support to individuals with HNC and factors of importance for their usage of and adherence to internet-based support remain to be explored. Systematic reviews suggest that interventions integrating multiple evidence-based modalities may increase ICs' preparedness for caregiving, decrease their caregiver burden and improve their health. Interventions must fulfil ICs' needs so they can provide the best possible patient care that their resources permit. Thus, internet-based interventions for ICs should be developed, tested and evaluated in collaboration with ICs and HCPs.

## AIMS

The main aim is to describe the development of an internet-based intervention (Carer eSupport) to improve ICs' ability to support individuals with HNC, and the testing of the Carer eSupport's feasibility and acceptability. An additional aim is to explore HCPs' perceptions

of the content and design of Carer eSupport from a clinical perspective.

## METHODS AND ANALYSIS
### Design
This protocol describes the planning of two phases. Phase I encompasses the development of Carer eSupport, based on focus groups discussion (FGDs) with ICs and HCPs, a theoretical framework and a literature review. ICs' and HCPs' perceptions of early versions of Carer eSupport will also be collected. Phase II is a one-arm feasibility study with a post-test design and includes individual interviews with ICs who will have had access to the first version of Carer eSupport for 1 month. This protocol complies with the Standard Protocol Items: Recommendations for Interventional Trials guidelines on writing protocols.[43]

### Setting and procedure phases I and II
Participants are recruited at ear, nose and throat and oncology and radiotherapy clinics at three Swedish university hospitals. Patients with HNC and their ICs will be identified by the HCPs responsible for patients care. ICs are designated by the patients, who must provide informed consent before their respective IC can be approached. HCPs who are asked to participate in the study will receive information about the study from a contact person at the hospital where they work, who will provide contact details to the research team (a project researcher). Next, ICs and HCPs will be contacted with more information about the study by the research team. They will be invited to participate and asked to provide informed consent.

### Phase I: development
#### Literature review and theoretical framework
A review of the literature on the complexity of being an IC to a patient with cancer, in particular a patient with HNC, has been conducted and is presented in the introduction above. The literature review findings, SCT[40] and UTAUT[42] will guide the development of a semi-structured interview guide for FGDs with ICs of patients with HNC.

#### Focus group discussions
##### Sample of ICs
ICs are designated by patients; their eligibility is assessed as follows:
Inclusion criteria
1. IC to a person with HNC about to start treatment, under treatment, or who has completed treatment not more than 3 months earlier.
2. Age >18 years.
Exclusion criteria
1. Confusion or cognitive impairments.
2. Unable to understand, speak, and read Swedish.

People of different genders and ages, with experience of being an IC to an individual with HNC, is planned to be invited—to form one or two groups of 6–8 ICs at each of the three study sites.

##### Sample of HCPs
HCPs are included to receive clinicians' aspects of internet-based support for ICs to facilitate an eventual implementation after the forthcoming RCT. The HCP groups will include people of different genders, ages and professions, for example, nurses, assistant nurses, physicians, psychologists, dietitians, speech therapists and dental hygienists. At least one group of 6–8 HCPs at each of the three study sites is planned to be included. The inclusion criteria is having experience of caring for patients with HNC.

##### The semistructured interviews
The FGDs with ICs[44 45] will be divided into two parts. The first will focus on ICs' perceived experiences and needs. This may include knowledge and skills related to caring for individuals with HNC and support that could decrease the caregiver burden and increase ICs' preparedness for caregiving and ability to consider their own health. Self-care strategies and the need for psychosocial counselling will be explored. During the FGDs, open questions will be used to address ICs' experiences, for example, 'What has been the major challenge for you as an IC to an individual with HNC?', 'What kind of support did you need as an IC to a person with HNC?', 'What support did you get that met your needs?' and 'What support did you feel was lacking in your role as an IC?'.

The second part will explore the ICs' positive and negative experiences of internet-based support and how it should be designed to have high utility and provide an optimal user experience. It will also explore ICs' thoughts on integration of AI into Carer eSupport, for example, whether it's content should be automatically adapted depending on how an IC uses it. If necessary, additional individual interviews will be performed.

The FGDs with HCPs will be used to understand how Carer eSupport should be designed from a clinical perspective, and thereby facilitate potential implementation in standard care. HCPs' perceptions of ICs' needs for support concerning caregiving and well-being, and their views on how such support can be delivered via the internet in a clinical setting will be explored. The interview guide for HCPs contains questions such as 'What support do ICs of individuals treated for HNC need?', 'Which professions are needed in the support of the ICs?', and 'How do you imagine that an internet-based support for ICs will be useful in clinical practice?'

Researchers (one facilitator and one moderator) and a PhD student will conduct the FGDs and notes will be taken by the moderator, to capture participants' nonverbal expressions.

The FGDs will be conducted online using end-to-end encryption Zoom software with assurance that no unauthorised person can enter the FGD. Zoom servers (San Jose, USA) in the European Union are used for Swedish

Universities and comply with General Data Protection Regulations[46] according to Swedish University regulations. The FGDs are expected to last 120–140 min, and will be audiotaped and transcribed.

## Analysis of the FGDs

Transcripts and notes from FGDs will be analysed using thematic analysis, research approach[47] to explore the underlying concepts and develop the intervention. By recording how codes will be developed from observations and ideas, the research team will use member checking and group debriefing during and after FDGs to support the trustworthiness of the researcher's interpretations and analysis.[48]

## Design and delivery of a first version of Carer eSupport

Carer eSupport will be provided via a web portal developed within the Uppsala University Psychosocial Care Programme (U-CARE).[49 50] The U-CARE portal (Portal) has been developed and refined since 2010 and has been used in several previous projects. The Portal is designed for building and delivery of interventions and include features for randomisation of study participants, collecting self-report data, sending reminders at specific times and logging user activity. The Portal uses a secure login process based on electronic identification, and all data are handled in a secure manner. A user-centred design methodology will be used in the development of the specific features of Carer eSupport. This implies that the U-CARE system developers take into account opinions and feedback from an IC expert group (see patient and public involvement) regarding the features of Carer eSupport repeatedly during the development process.[51] The content of Carer eSupport will be based on the best available scientific evidence and the results of the FGD. The IC expert group will provide opinions regarding the content at several meetings throughout the development of the content. The research team has extensive knowledge of development, testing and evaluation of internet-based support, including usability and user experience. They will have the main responsibility for the contents and the development of Carer eSupport, in collaboration with the IC expert group and HCP experts. HCPs ensure that the entire spectrum of ICs' needs is considered.

## The HCPs' perceptions of an early version of Carer eSupport

The HCPs will be given access to an early version of Carer eSupport for 1 month. This is done to receive their view about how Carer eSupport may be improved before the feasibility testing. Individual interviews will be held with them regarding the features of Carer eSupport, its content and design, and if they encountered any technical problems. The interviews will be conducted online by the research team, using end-to-end encryption Zoom software (San Jose, USA).

## Phase II: feasibility testing

Phase II encompasses testing the relevance, acceptability, and user experience of Carer eSupport.[4]

## Sample of ICs

The recommendations of Cocks et al on sample size in feasibility studies[52] would mean that approximately 30 ICs of patients with HNC need to be included. The feasibility and acceptability of Carer eSupport will be evaluated to increase the robustness of the forthcoming RCT.

Inclusion criteria
1. IC of an individual with HNC who is about to start, is undergoing, or has completed primary oncology treatment not more than 1 month earlier.
2. Age>18 years.

Exclusion criteria
1. Confusion or dementia.
2. Unable to understand, speak, and read Swedish.

## The test version of Carer eSupport

This is a tentative description of the first version of Carer eSupport which still is to be developed during phase I. ICs will be given access to Carer eSupport for 1 month. The support may be delivered in the form of electronic live education for ICs and a library with written and audiovisual material. It may be possible to send messages and have conversations via a video link with HCPs, use discussion forums and use digital rooms to exchange experiences with other ICs in a similar situation.

## Feasibility testing
### Explanation and justification of method

This feasibility study will use a mixed-method design, including analysis and integration of qualitative and quantitative data.[44] In the qualitative approach, individual semistructured interviews with ICs will be conducted after the test period. The interviews will explore the ICs' experiences of Carer eSupport (eg, the content, functions, layout and technical problems). Their experiences with the questionnaires, the recruitment to the study will also be explored. The individual interviews will be conducted face to face or online by the researchers, using end-to-end encryption Zoom software (San Jose, USA). In the quantitative approach, log data from Carer eSupport will be collected after the test period to examine the ICs' activities and use patterns in the system, and to identify any indications of technical problems. This data will provide knowledge about how Carer eSupport, the recruitment procedures and the choice of outcomes may be improved to increase the robustness of the forthcoming RCT. Depending on the results of the feasibility study it may be relevant to make changes in the Carer eSuppoprt.

Carer eSupport will be considered feasible if ≥45% of eligible, approached ICs are recruited in participation, ≥50% of included ICs use the support (based on number of logins, use of educational materials according to log data and individual interviews), attrition is <30% and ≥50% complete the questionnaires.[52] In addition, to be deemed feasible, Carer eSupport must be considered acceptable, relevant and usable, based on qualitative data.

## Questionnaires

The ICs will be asked to complete questionnaires via the Portal (see below) before they are given access to Carer eSupport. These questionnaires are planned to be used in a RCT. However, questionnaires may be added or removed, depending on the outcome of phase I and the feasibility testing.

*The Preparedness for Caregiving scale*[53 54] is used to measure caregivers' perceived readiness to provide care in real time. It consists of eight items measuring preparedness to take care of the patient's needs, to obtain and set up services, to get necessary support from the healthcare system and to manage IC-related stress. The scores are added to a total score ranging from 0 to 32, with a higher score reflecting better preparedness.

*The Caregiver's Burden scale*[55] will be used to assess IC burden. It consists of 22 items covering areas known to be especially important, such as IC health, psychological well-being, relationships, social network and physical workload. Items are scored on a 4-point Likert scale, with higher scores indicating more burden.

*The Short Form 36 version 2 (SF-36v2)* is used to measure self-perceived health. It has been shown to have good psychometric properties.[56] It measures physical functioning, role limitations, bodily pain, general health, vitality, social functioning and mental health. SF-36v2 encompasses a physical component and a mental component summary.

*The Depression Anxiety Stress Scale-21 (DASS-21)*[57] comprises three 7-item subscales measuring depression, anxiety, and stress. Each item is scored on a 4-point Likert scale, with higher scores indicating more symptoms. The DASS-21 has good psychometric properties.[58]

*The Multidimensional Fatigue Inventory (MFI-20)*[59] will be used to measure fatigue. The questionnaire consists of 20 items that assess five dimensions of fatigue. A total score, ranging from 4 to 20, is calculated for each dimension by summation of the individual item scores. The Swedish version has good psychometric properties, for patients and healthy individuals.[60]

*Clinical and sociodemographic data* will be collected from ICs, including age, gender, marital status, relationship to the patient, children living at home, level of education, psychological support, employment, economic situation and computer skills. Data about patients' age, gender, diagnosis and tumour stage and treatment (surgery, radio therapy, and pharmacological treatment) will be collected from medical records.

During the individual interviews (described above), the ICs will be asked about their experiences of the questionnaires, for example, 'What was your experience of answering the questionnaires?', 'Did you think the questions were relevant to answer initially and after a month?', 'Did you find the questions easy to understand' and 'Do you have thoughts and feelings that we haven't asked you about?'

## Analysis

The quantitative data, for example, demographic data and log data, will be analysed using descriptive statistics. The individual interviews will be analysed using qualitative content analysis following the approach of Graneheim and Lundman.[61] To confirm the rigourousness of the analysis, the researchers will document and write extensive and detailed notes of the emergent analytical and theoretical insights.

## Time plan

The preliminary timing of phase I is April 2021–March 2022, with phase II in March–November 2022.

## The research team

Our research group was formed in 2020 and consists of HCPs from various fields and two experts in human–computer interaction. The members in the group have extensive experience of developing, testing and evaluating internet-based complex interventions for ICs and patients.[62–65] Our group includes expertise in the field of human–computer interaction and user-centred design,[66] especially eHealth applications, including gender aspects,[67] expertise regarding ICs in the cancer context[68] and expertise in nursing and medical aspects of patients with HNC.[69]

## Patient and public involvement

One of the researcher in the research team has own experience of being an IC to an individual treated for HNC. In addition we will form an expert panel with ICs to regularly advise the research group about the content and design of Carer eSupport and how the project should be executed. Further, we will invite an advisory board, consisting of researchers, managers in healthcare, politicians and patient organisation representatives to seminars, who can regularly provide their opinions on the project's progress and clinical relevance at seminars.

## DISCUSSION

Carer eSupport will be developed based on the experiences of ICs and HCPs, scientific evidence, SCT and UTAUT. This will add to the theoretical basis regarding ICs' needs. The goal is to develop a feasible and acceptable internet-based intervention to be evaluated in a forthcoming RCT.

We will perform a thorough review, as recommended by O'Cathain *et al*[6] instead of following a systematic review methodology. Undertaking a systematic review is not always necessary when developing a complex intervention, since there may be recent reviews available—nor is it always possible, given limited resources. However, this may be a limitation, as we may overlook valuable information from past research.

A thorough conduction of phases I and II and establishment of an expert panel consisting of ICs and an advisory group, providing input on how the project can be carried

out, will be crucial for the relevance of Carer eSupport. If the forthcoming RCT shows Carer eSupport to be effective regarding ICs' preparedness for caregiving and IC health, this will increase the probability that Carer eSupport can be implemented into healthcare.

Carer eSupport will be adapted to ICs who need to be prepared for caregiving at home due to the advanced treatment given to the individuals they provide care to.[11 12] Engaging stakeholders, for example, end users, in the development phase[6 7] is therefore necessary to meet the ICs' needs. These stakeholders must have experience of caring for individuals with HNC at home, so they can find solutions that may make a difference for the ICs and identify and fill the gaps in relation to their needs. This is in line with the suggestions of Slev *et al*[70]: that future studies should develop interventions and provide insight into the effects of eHealth on ICs in particular.

If a researcher does not perform a process evaluation when implementing a complex intervention, it is assumed that the implementation will proceed correctly, which rarely occurs. Therefore, we will perform a process evaluation. However, given the limited sample size, conclusions drawn should be considered with caution and supplemented with a process evaluation of the future RCT.

We are aware of the difficulties that may arise of conducting FGDs online. Some ICs may be unable to participate due to technical difficulties or not owning a computer. This may result in a lack of certain groups' experiences and needs. Also, patients with HNC make up a heterogenous group with differing needs and prognosis. We acknowledge this as a limitation. The intervention is adapted for ICs of individuals with HNC who have ongoing treatment or have been treated, and it might be a challenge to meet all their needs and expectations.

We believe that ICs of individuals with HNC are motivated to learn caregiving skills.[71] Previous research has shown that ICs experience lack of preparedness. Carer eSupport may provide ICs with support and opportunities to acquire new skills and increase their self-efficacy for caregiving. SCT[40] provides a useful framework for understanding the ICs' behaviours and needs.[41] According to SCT, self-efficacy and behavioural capabilities, such as perceived preparedness, as well as environmental factors, may be important predictors of IC distress and could be useful for identifying those who might benefit most from Carer eSupport. A strength is that we have complemented the theoretical framework with UTAUT,[42] to understand how each individual accepts technology. These theories serve as complementary tools for us to analyse the results and understand how to build Carer eSupport.

**Author affiliations**
[1]Department of Immunology, Genetics and Pathology, Experimental and Clinical Oncology, Uppsala University, Uppsala, Sweden
[2]Department of Information Technology, Uppsala University, Uppsala, Sweden
[3]Department of Public Health and Caring Sciences, Uppsala University, Uppsala, Sweden
[4]Department of Women's and Children's Health, Healthcare Sciences and e-Health, Uppsala University, Uppsala, Sweden
[5]Department of Surgical Sciences, Section of Otorhinolaryngology and Head and Neck Surgery, Uppsala University, Uppsala, Sweden

**Twitter** Åsa Cajander @AsaC

**Contributors** BJ conceived the study, contextualised Carer eSupport, developed the protocol, is the grant holder of this study, and contributed critical input during drafting and finalisation of the manuscript. All authors provided input on the design of Carer eSupport. UL drafted and finalised the manuscript. YTE contributed with critical input during drafting and finalisation of the manuscript. ÅC, MC and AA provided critical input to finalise the protocol. GL and LvE contributed by critically commenting on and approving the final manuscript.

**Funding** This work was supported by the Swedish Research Council (grant number 2019-01231), the Swedish Cancer Society (grant number 20 1014 PjF), the Oncology Department Foundations Research Fund in Uppsala (grant number N/a) and the Swedish state under the agreement between the Swedish government and the county councils, the ALF- agreement (Uppsala county: ALF-941900).

**Competing interests** None declared.

**Patient and public involvement** Patients and/or the public were involved in the design, or conduct, or reporting or dissemination plans of this research. Refer to the Methods section for further details.

**Patient consent for publication** Consent obtained directly from patient(s).

**Provenance and peer review** Not commissioned; externally peer reviewed.

**ORCID iDs**
Ulrica Langegård http://orcid.org/0000-0001-8174-579X
Åsa Cajander http://orcid.org/0000-0001-7472-2215
Göran Laurell http://orcid.org/0000-0002-7760-246X
Ylva Tiblom Ehrsson http://orcid.org/0000-0001-7435-167X

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
