## [Reviewer comments · BMJ Open]

ARTICLE DETAILS

TITLE (PROVISIONAL)	Internet based support for informal caregivers to individuals with head and neck cancer (Carer eSupport): A study protocol for the development and feasibility testing of a complex online intervention
AUTHORS	Langedård, Ulrica; Cajander, Åsa; Carlsson, Maria; von Essen, Louise; Ahmad, Awais; Laurell, Göran; Tiblom Ehrsson, Ylva; Johansson, Birgitta

VERSION 1 – REVIEW

REVIEWER	Krishnasamy, Meinir University of Melbourne, Nursing
REVIEW RETURNED	24-Oct-2021

GENERAL COMMENTS	This is an interesting study, recognising unmet need of informal carers of people diagnosed with head and neck cancers. This protocol paper describes design and intent for Phases 1 and 2 of a program of work, to develop and test a digital resource to support informal carers. Commentary is offered below to further strengthen the manuscript. I appreciate that some work may have commenced and as such it may not be possible to address all questions raised below. Edits/typos Line 1 in the abstract: replace may, with often or can. Throughout the manuscript there is reference to ICs “to” people with H&N cancers. Suggest this is changed throughout to ICs “of” people with ... Page 7. Last paragraph; line 1- change “giving” to ‘providing’. Abstract Para 1: Aims. The aims could be strengthened by re-wording /clarifying focus. For example, rather than “develop a user friendly internet based intervention for ICs of ...and test its feasibility and acceptability”, the authors could consider “Develop and test clinical utility of an internet based.....”
--

As a concept, clinical utility includes constructs broader than feasibility & acceptability, and this would better align with the focus of aims as outlined in the manuscript e.g., page 8 research questions include relevance, acceptability, utility, user-experience. Smart's paper on multi-dimensional clinical utility offers a good framework for building a clinical utility study and would help strengthen the data analysis section on page 14, which currently does not outline any detail or plan for how each of the components of feasibility etc are to be measured. (Smart A. A multi-dimensional model of clinical utility. Int J Qual Health Care. 2006 Oct;18(5):377-82).

P. 12. Feasibility testing is described as recruitment, questionnaire completion, use of resources. These are appropriate feasibility criterion for an RCT, but these are not comprehensive criterion for establishing acceptability, patient centredness, etc. The purpose of the phases 1 and 2 of the study are to establish content relevance, ease of use of the resource etc. These data may result in changes to the resource that may ultimately change/improve recruitment/retention rates etc. And so, if these data are collected in phase 2, they may give a false impression of whether people will engage with and be able to apply interventions included the resource, etc.

I appreciate that the interviews will be undertaken to explore experience of relevance, usability etc, but organising each of these different components and the aligned data collection approaches within a clinical utility framework would strengthen the protocol. The Smart paper offers a robust and pragmatic table to show how this can be done.

If this is changed, the article summary will need to be revised.

Introduction

I appreciate that the authors intent to conduct a review of the literature as part of phase 1, but there are key papers missing from the introduction that would help further strengthen the case for this study. For example, the Aung and Wang papers cited below. The authors could include these as part of their review of the manuscript.

References 6-7 and 9-10 are focused on nutrition and coping strategies related to food. Is this a particular focus of the work? A broader perspective on need experienced by patients would be more appropriate for a broad background/introduction e.g.

Wells M, Cunningham M, Lang H, Swartzman S, Philp J, Taylor L, Thomson J. Distress, concerns and unmet needs in survivors of head and neck cancer: a cross-sectional survey. *Eur J Cancer Care (Engl)*. 2015 Sep;24(5):748-60.

In the introduction with regard to reference 11, this is an important point, but there is no context to explain the relevance of this. Similarly for references 12 and 13. These are presented as statements but no explanation of the impact of these is made - i.e., why is addressing these needs is important; e.g., diminished capacity to care for patients; long-term carer morbidity (e.g., ref 12 followed people out to 6 months only); health service utilisation; preferred place of death attainment etc. Why is investment in caring for ICs important? I appreciate this seems an obvious point - but stating this clearly will strengthen the rationale and need for your work.

Importantly, there is no reference to the work of:

D'Souza V, Blouin E, Zeitouni A, Muller K, Allison PJ. Multimedia information intervention and its benefits in partners of the head and neck cancer patients. *Eur J Cancer Care (Engl)*. 2017 Jul;26(4). doi: 10.1111/ecc.12440. Epub 2016 Jan 18. PMID: 26777257;

or,

Slev VN, Mistiaen P, Pasman HR, Verdonck-de Leeuw IM, van Uden-Kraan CF, Francke AL. Effects of eHealth for patients and informal caregivers confronted with cancer: A meta-review. *Int J Med Inform*. 2016 Mar;87:54-67, who did not find any ehealth resources for ICs

Literature not referenced regarding the needs of ICs include:

Wang T, Mazanec SR, Voss JG. Needs of Informal Caregivers of Patients With Head and Neck Cancer: A Systematic Review. *Oncol Nurs Forum*. 2021 Jan 4;48(1):11-29. doi: 10.1188/21.ONF.11-29. PMID: 33337440.

Halkett GK, Golding RM, Langbecker D, White R, Jackson M, Kernutt E, O'Connor M. From the carer's mouth: A

phenomenological exploration of carer experiences with head and neck cancer patients. *Psychooncology*. 2020 Oct;29(10):1695-1703. doi:

Aung SHH, White K, Bloomfield J. The Experiences and the Needs of Caregivers of Patients With Head and Neck Cancer: An Integrative Review. *Cancer Nurs*. 2020 Jun 12. doi: 10.1097/NCC.0000000000000833. Epub ahead of print. PMID: 32541207.

Schaller A, Liedberg GM, Larsson B. How relatives of patients with head and neck cancer experience pain, disease progression and treatment: a qualitative interview study. *Eur J Oncol Nurs*. 2014 Aug;18(4):405-10. doi: 10.1016/j.ejon.2014.03.008. Epub 2014

Fronczek AE. A Phenomenologic Study of Family Caregivers of Patients With Head and Neck Cancers. *Oncol Nurs Forum*. 2015 Nov;42(6):593-600. doi: 10.1188/15.ONF.593-600. PMID: 26488829.

Internet-based interventions

On page 6, the section starts with the sentence, "Internet based interventions need to be further studied..." What is the evidence for this? If phase 1 of the study is about developing a new resource, should the mode of delivery of this resource be part of that discussion? In Richardson's paper (ICs stated that their preference was for face to face support – this may be an artefact of when the study was done) but this is an important component of resource utility or acceptability/person-centredness.

Richardson AE, Morton R, Broadbent E. Caregivers' Illness Perceptions Contribute to Quality of Life in Head and Neck Cancer Patients at Diagnosis. *J Psychosoc Oncol*. 2015;33(4):414-32.

On the top of p. 7 there is reference to AI but as above, given the approach to Phase 1, should this kind of feature or component be identified during the interviews, rather than be pre-empted?

Theoretical Framework

Does inclusion of the UTAUT presuppose that ICs want or need a technological solution? Strengthening the rationale for this assumption in the introduction will help address this question.

	Aims Is the resource for all stages of H&N cancer care? Will the resource be targeted to ICs at time of diagnosis; during or into survivorship? Will it address endo of life issues? The literature suggests that support needs of patients with H&N Caners and their ICs change over time with informational needs declining while emotional needs persist longer and change depending on complexity of care needs and tasks; resilience of ICs, etc. Why explore health professionals' perception of content and design? Is this so they will recommend it to their patients? Research questions need to be structured around the intent of the project eg how feasible is; how acceptable is; what content do ICs of people with H&N cancers say they need; how do they want this information etc . The questions could be grouped around the components of clinical utility and then the data collection and analysis approaches can be structured around these components. The questions assume an IT solution – what if the interviews show that ICs prefer face to face? Design There is reference to the MRC framework which is about stages of intervention development and it suggests design approaches, but no clear methodological design is presented for this study. For example, an experience-based co-design methodology could be applied to phase 1 (e.g. IC and HCP perspectives collected separately and then brought together; why not include patients, they will be able to inform what support they need form ICs and so help target the skills list that ICs may need to develop capability in to reduce burden, concern etc). At the top of p. 11 there is reference to analysis of the qualitative data. This needs more detail. E.g., what kind of content analysis will be used – latent/manifest etc? Who will the member checking be undertaken with/when? What does group debriefing refer to and what is meant by emergent analytical and theoretical insights. If the interviews are being undertaken to develop or test theory, this needs to be explicit and explained in the content of the purpose of the first phase. The purpose of phase 2 isn't clear. On page 11 it states that it's about testing different utility domains, but a "one arm feasibility post-test is presented as the design". It phase 2 about establishing
--	--

utility domains of the new resource before its potential for impact is pilot-tested ahead of an RCT? What is the purpose of the measures? Is it to test their applicability for a future pre-post pilot test and future RCT? The study proposed for phase 2 will be too small to carry out any test of potential to benefit.

As stated above, applying RCT feasibility criteria does not seem appropriate here. This would be more meaningfully done once the resource has been finalised and is ready for pilot testing.

Questionnaires

In the introduction, there is reference to fear of cancer recurrence as being a issue for ICs. There is no measure included to address this? If true to the methodology of Phase 1 – especially if this is about experience-based co-design or co-production, the measures to be included should be decided after analysis of the phase 1 focus groups, ensuring they target concerns importance/relevance to ICs.

Analysis

More detail is required to explain what descriptive statistics will be applied and to what data. Again, using the utility framework will help clearly articulate this

What is meant by “additional tests will be conducted if large scale changes are made” ? What kinds of tests and what/how is a large-scale change (and to what) defined? Some of these issues may not be possible to define until after Phase 1 has been undertaken.

Phase 1 will generate data to build the prototype (content/structure etc). What is the process from generating qualitative insights to building a prototype that can be tested for utility in phase 2?

Discussion

Paragraph 4. This is a long sentence that needs restructuring to be clear. Will an implementation science framework be used to guide a future RCT- eg hybrid implementation trial?

Paragraph 5 – this is the first mention of tailoring the intervention to high burden ICs. Is this what the resource is intended for? If so this needs to be made clear from the outset, and will impact recruitment to phases 1 and 2- e.g. would people have to complete

	self-report measures of level of perceived burden/need to be eligible to take part in phases 1 and 2 of the study? What is meant by “find solutions that make a significant difference in system development”? Is system the resource and how will significance be measured? Depending on authors’ responses to the comments above, Figure 1 will need revising. References Some references are incomplete. Please check the journal reference style/requirements.
--	---

REVIEWER	Arends, Jann Universitätsklinikum Freiburg, Klinik für Innere Medizin I
REVIEW RETURNED	08-Nov-2021

GENERAL COMMENTS	The planned multi-centered study is aimed at developing and testing a web-based interaction tool to support non-professional carers of patients with cancers of the head and neck region. The design of the study appears reasonable; the expertise available for all aspects of the planned project appears to be of high quality and thus adequate to perform this study; the time-frame to recruit patients and to develop and test the web-based tool also appears reasonable. I wish good success for this trial.
--

VERSION 1 – AUTHOR RESPONSE

Reviewer I	
Edits/typos Line 1 in the abstract: replace may, with often or can.	√
Throughout the manuscript there is reference to ICs “to” people with H&N cancers. Suggest this is changed throughout to ICs “of” people with	√
Page 7. Last paragraph; line 1- change “giving” to ‘providing”.	√
Abstract #1. Para 1: Aims. The aims could be strengthened by re-wording /clarifying focus. For example, rather than “develop a user friendly internet based intervention for ICs of ...and test its feasibility and acceptability”, the authors could	Please see the revised aim, the explanation in #2 and the added text in page 5.

consider “Develop and test clinical utility of an internet based	
#2. As a concept, clinical utility includes constructs broader than feasibility & acceptability, and this would better align with the focus of aims as outlined in the manuscript e.g., page 8 research questions include relevance, acceptability, utility, user-experience. Smart’s paper on multidimensional clinical utility offers a good framework for building a clinical utility study and would help strengthen the data analysis section on page 14, which currently does not outline any detail or plan for how each of the components of feasibility etc are to be measured. Feasibility testing is described as recruitment, questionnaire completion, use of resources. These are appropriate feasibility criterion for an RCT, but these are not comprehensive criterion for establishing acceptability, patient centredness, etc. The purpose of the phases 1 and 2 of the study are to establish content relevance, ease of use of the resource etc. These data may result in changes to the resource that may ultimately change/improve recruitment/retention rates etc. And so, if these data are collected in phase 2, they may give a false impression of whether people will engage with and be able to apply interventions included the resource, etc. I appreciate that the interviews will be undertaken to explore experience of relevance, usability etc, but organising each of these different components and the aligned data collection approaches within a clinical utility framework would strengthen the protocol. The Smart paper offers a robust and pragmatic table to show how this can be done.	Thank you, we appreciate your suggestions. We understand that you want a clarification. Respectfully, we want to explain the ongoing work and what we are planning to do in the future. We have added a paragraph in the introduction part (page 5) where we describe and clarify that we will perform a process evaluation and use two complementary models, Precede-Proceed model and (PPM) and The DeLone and McLean Model. We believe that PPM will serve and organize the framework, clarify, and guide the selection of research questions and methods. PPM allows leeway to adapt the content and methods of the intervention to the particular needs and circumstances. Thus, we see some shortcomings in PPM related to the implementation of IT specifically, so we supplement with an additional model, The DeLone and McLean Information Systems Success Model that include the concepts we use which are more closely connected to the IT system such as information quality, service quality, intention to use and user satisfaction.
Introduction #3. I appreciate that the authors intent to conduct a review of the literature as part of phase 1, but there are key papers missing from the introduction that would help further strengthen the case for this study. For example, the Aung and Wang papers cited below. The authors could include these as part of their review of the manuscript.	Thank you for a very good suggestion. We added five of your suggested references.  - Wang T, et al. Oncol Nurs Forum. 2021 - Slev VN, et al. Int J Med Inform. 2016. - D'Souza V, et al. Eur J Cancer Care. 2017. - Aung SHH, et al. An Integrative Review. Cancer Nurs. 2020. - Wells M, et al. Eur J Cancer Care. 2015
#4. References 6-7 and 9-10 are focused on nutrition and coping strategies related to food. Is this a particular focus of the work? A broader	Please see the revised text, page 5 lines 118-126.

perspective on need experienced by patients would be more appropriate for a broad background/introduction e.g.	
#5. In the introduction with regard to reference 11, this is an important point, but there is no context to explain the relevance of this. Similarly for references 12 and 13. These are presented as statements but no explanation of the impact of these is made - i.e., why is addressing these needs is important; e.g., diminished capacity to care for patients; long -term carer morbidity (e.g., ref 12 followed people out to 6 months only); health service utilisation; preferred place of death attainment etc. Why is investment in caring for ICs important? I appreciate this seems an obvious point - but stating this clearly will strengthen the rationale and need for your work.	We agree and have added information why investment in caring for ICs is important. Please see the revised text on page 6 “Preparedness and caregiver burden”
Internet-based interventions #6. On page 6, the section starts with the sentence, “Internet based interventions need to be further studied...” What is the evidence for this? If phase 1 of the study is about developing a new resource, should the mode of delivery of this resource be part of that discussion? In Richardson’s paper (ICs stated that their preference was for face to face support – this may be an artefact of when the study was done) but this is an important component of resource utility or acceptability/personcentredness.	Our goal is that this internet-based health care interventions will be a complement to standard care, not replace the face to face meetings with the health care professions. Please see the revised text on page 7 line 170-173.
#7 On the top of p. 7 there is reference to AI but as above, given the approach to Phase 1, should this kind of feature or component be identified during the interviews, rather than be pre-empted?	Please see the revised text on page 8 lines 188-189.
Theoretical Framework #8. Does inclusion of the UTAUT presuppose that ICs want or need a technological solution? Strengthening the rationale for this assumption in the introduction will help address this question.	The UTAUT model can help to understand the use and non-use of a technical solution. This does not presuppose that ICs want or need a technical solution. We have added an explanation in the introduction. Page 8 line 205. It is important that it appears that this intervention is a complement to the standard care, an intervention that is available 24 hours a day, when it fits the ICs, or ICs to patients who are treated far from home. It should act as a technical solution.
AIM #9. Is the resource for all stages of H&N cancer care? Will the resource be targeted to ICs at time of diagnosis; during or into survivorship? Will it address end of life issues? The literature suggests that support needs of patients with	We agree that the literature indicates that support is needed for patients with all stages of HNC and also their IC and that their needs change over time. During the upcoming RCT, we will collect data via the portal at baseline,

H&N Cancers and their ICs change over time with informational needs declining while emotional needs persist longer and change depending on complexity of care needs and tasks; resilience of ICs, etc.	post-intervention, and at 3 months follow-up that will cover the acute side effects of radiation therapy. We are focused on the primary oncology treatment period in phase 2. Please see the revised inclusion criteria on page 12. We expect that the upcoming RCT will show that ICs using Carer eSupport will experience a higher degree of preparedness for caregiving at home with a reduced care burden, which may lead to increased health over time for both ICs and patients.
#10. Why explore health professionals' perception of content and design? Is this so they will recommend it to their patients?	Focus groups interviews with healthcare professionals responsible for the care of individuals with HNC and support to ICs, will be made to develop an understanding of how Carer-eSupport should be designed to be usable from a clinical perspective and thereby facilitate implementation in standard care, given that Carer eSupport is effective according to the RCT. The reason why we ask the healthcare staff for participating in the study is that the implementation of a complex intervention presupposes that the staff is involved and contributes with their views and knowledge. Without healthcare staff, it will not be useful, although parts of the research group have extensive clinical experience in head and neck cancer care, we need expert advice and experience from different health care professionals. We expect is that the intervention will be clinically useful in the future, therefore the staff's views are necessary to include from the start. See pages 11 line 289.
#11. Research questions need to be structured around the intent of the project eg how feasible is; how acceptable is; what content do ICs of people with H&N cancers say they need; how do they want this information etc. The questions could be grouped around the components of clinical utility and then the data collection and analysis approaches can be structured around these components.	We agree that the aim needs to be reformulated. Please see the revised aim on page 9.
#12. The questions assume an IT solution – what if the interviews show that ICs prefer face to face?	This intervention is a complement to the health care, not a replacement. Please see the answer at#8.

Design #13. There is reference to the MRC framework which is about stages of intervention development and it suggests design approaches, but no clear methodological design is presented for this study. For example, an experience-based co-design methodology could be applied to phase 1 (e.g. IC and HCP perspectives collected separately and then brought together; why not include patients, they will be able to inform what support they need from ICs and so help target the skills list that ICs may need to develop capability in to reduce burden, concern etc).	Please see the revised text on page 9 lines 233-240. Thank you for a very reflective statement. Since Carer eSupport should be adapted to ICs need we have chosen not to include the patients view about how the intervention should be designed. The patients and ICs may have different opinions and experiences which will risking blurring the focus of ICs needs.
#14. At the top of p. 11 there is reference to analysis of the qualitative data. This needs more detail. E.g., what kind of content analysis will be used – latent/manifest etc? Who will the member checking be undertaken with/when? What does group debriefing refer to and what is meant by emergent analytical and theoretical insights. If the interviews are being undertaken to develop or test theory, this needs to be explicit and explained in the content of the purpose of the first phase.	We have specified the method we will use to analyze the focus groups discussion, page 12 line 306. According to your comment about latent/manifest analysis: Since a thematic analysis refers to themes, the notion of a theme must be examined more closely. Specific criteria need to be stipulated concerning what can and cannot be coded within such themes; otherwise this form of content is highly subjective. Thematic analyses tend to draw on both types of themes. Often one can identify a set of manifest themes, which point to a more latent level of meaning. The deduction of latent meanings underpinning sets of manifest themes requires interpretation. Thus, we wait until we perform the analysis.
#15. The purpose of phase 2 isn't clear. On page 11 it states that it's about testing different utility domains, but a "one arm feasibility post-test is presented as the design". It phase 2 about establishing utility domains of the new resource before its potential for impact is pilot-tested ahead of an RCT? What is the purpose of the measures? Is it to test their applicability for a future pre-post pilot test and future RCT? The study proposed for phase 2 will be too small to carry out any test of potential to benefit. As stated above, applying RCT feasibility criteria does not seem appropriate here. This would be more meaningfully done once the resource has been finalised and is ready for pilot testing.	Thank you, this was not clearly described of the former version of the manuscript. The questionnaires are included to test their applicability for a future RCT and not for evaluation of the effects of Carer eSupport. We have clarified this in page 13, lines 378-381. The aim of phase 2 is to test the feasibility of Carer eSupport in a group of ICs and with no control group (one-arm). We do not have a baseline for the feasibility testing, only quantitative and qualitative data after the test have been completed (post-test). This has been clarified in page 13, line 362.
Questionnaires #16. In the introduction, there is reference to fear of cancer recurrence as being a issue for ICs. There is no measure included to address this? If	The choice of the variables is based on the literature review. These questionnaires are planned to be used

true to the methodology of Phase 1 – especially if this is about experience-based co-design or co-production, the measures to be included should be decided after analysis of the phase 1 focus groups, ensuring they target concerns importance/relevance to ICs.	in the forthcoming randomized controlled trial. But, questionnaires may be added or withdrawn depending on the results of phase I and II. Please see the revised text in page 14. Lines 378-381 and page 15 line 422.
Analysis #17. More detail is required to explain what descriptive statistics will be applied and to what data. Again, using the utility framework will help clearly articulate this	See revised text page 15 line 425.
#18. What is meant by “additional tests will be conducted if large scale changes are made” ? What kinds of tests and what/how is a large-scale change (and to what) defined? Some of these issues may not be possible to define until after Phase 1 has been undertaken.	See revised text page 15 line 422.
#19. Phase 1 will generate data to build the prototype (content/structure etc). What is the process from generating qualitative insights to building a prototype that can be tested for utility in phase 2?	Through the literature, clinical expertise, the results from the focus group, ie a number of themes, will be concretized into supportive efforts (content in carer-esupport) and accordingly build modules in the portal with additional support from ICs in the expert group. Page 12 lines 320-324.
Discussion #20. Paragraph 4. This is a long sentence that needs restructuring to be clear. Will an implementation science framework be used to guide a future RCT- eg hybrid implementation trial?	Please see the revised text. As we have described earlier we will use MRC guidelines and will work thoroughly with the logical model PPM. MRCs recommendations is to start already during the feasibility study to learn the method and will encounter obstacles already there. We will be well prepared.
#21. Paragraph 5 – this is the first mention of tailoring the intervention to high burden ICs. Is this what the resource is intended for? If so this needs to be made clear from the outset, and will impact recruitment to phases 1 and 2- e.g. would people have to complete self-report measures of level of perceived burden/need to be eligible to take part in phases 1 and 2 of the study?	We have clarified and rewritten the text, page 17.
#22. What is meant by “find solutions that make a significant difference in system development”? Is system the resource and how will significance be measured?	We have clarified the text, page 17.
#23. Depending on authors’ responses to the comments above, Figure 1 will need revising.	With great respect, since we have chosen to keep the design in its origin no changes have been made to the figure, but we have changed the manuscript according to your suggestions.
#24. Some references are incomplete. Please check the journal reference style/requirements	√

	Thank you very much for your thorough review of our manuscript. We intent to present Phase I and II of MRC guidelines. Phase III will present the RCT and we will also conduct a process evaluation. Your comments have made us aware that we need to clarify several parts. You are right that some parts are already ongoing and are difficult to change. Our research group consists of experts from the human computer field who believe that the Preced-Proceed model together with The DeLone and McLean Model supports our work. We hope we have clarified this and that you understand how we think and accept our proposal.
Reviewer II	We thank you for reviewing our manuscript. Your positive opinion was very encouraging.

VERSION 2 – REVIEW

REVIEWER	Krishnasamy, Meimir University of Melbourne, Nursing
REVIEW RETURNED	13-Feb-2022

GENERAL COMMENTS	Thank you for the opportunity to review this paper that addresses an important area regarding development of internet based cancer supportive care for informal carers of people affected by head and neck cancers. The team are to be commended for their work. In the PDF attached below I have provided detailed feedback, with request for clarification of the study methodology. In particular, clearly defining what is meant by feasibility will greatly help clarify the intent of the work. For example, is this about clinical utility- how feasible is this to do in practice; or is it a pilot study to establish feasibility of undertaking an RCT. At times the distinction between both is unclear. No research questions are set and so it is unclear what data are being collected to address what aspects of feasibility; and no feasibility framework is cited (e.g., Proctor or Peters) with rationale for what aspects of feasibility are to be tested and why they are relevant to the study aims. There is reference to user-centred design but the ICs and HCPs seem to working separately in phase 1, with IC decisions about intervention development being checked or refined by HCPs. There is reference to phase 2 using quantitative and qualitative methods but no reference to a mixed methods design and the detail provided for analysis of phase 2 is very limited. There is opportunity to clarify rationale for the methodology that will strengthen the paper. I hope the comments in the manuscript are helpful.
--

VERSION 2 – AUTHOR RESPONSE

Reviewer I	Many thanks for the careful review of our manuscript and for your encouraging words.
# 0 The title could be further clarified - e.g, is this a study to develop and pilot test feasibility of undertaking an RCT, or an implementation science study where the paper is about development and establishing feasibility of implementation of a new intervention?	We chose not to rephrase the title but to add a clarification of why phases I and II will be conducted, at the very beginning of the introduction.
#1 Suggest: transition of some aspects of care; rather than suggesting care in its totality	Please see the revised text on page 3. However, we do not want to specify “at home or after discharge from the hospital”, we want to include the ICs from the beginning at diagnosis.
#2 The aims for this protocol need to be separated from the future study	We agree that we need to clarify our aim and separate the future study. Please see the revised text on page 3.
#3 Suggest changing usable to acceptable or feasible to retain consistency with language. It may be usable but not acceptable etc.	Please see the revised text on page 3.
#4 Think there is a word missing here	√
#5 Therefore, is this a mixed methods design? If so, this isn't articulated in the methods	FGDs and interviews in phase I will be analyzed using a qualitative method. In phase II, we will use both qualitative and quantitative methods in the same study. Please see the clarified text in the method, page 13.
#6 Should this be the study is the first to, rather than the protocol is the first to?	Please see the revised text on page 4.

#7 If this study is using user-centred design, the language is important	Please see the revised text on page 4.
# 8 As above- was this a mixed methods design?	Please see the revised text on page 4.
#9 Please clarify what is meant by a solid review	We have removed the word solid and rewritten the point, on page 4
#10 As the development phase is guided by user co-design principles, what happens if ICs say they do not want the intervention delivered via Carer eSupport?	We understand what you mean. We have clarified that the portal has also been developed in collaboration with ICs. This reduces the risk that users will not use the portal. Page 4
# 11 This sentence needs a reference.	√. Page 5
# 12 Please clarify what is meant by framing.	Please see the revised text on page 5
# 13 Given these sentences, should an implementation science framework be used?	We have clarified that it is in the future the implementation is provided that the intervention is effective, page 5
# 14 If the paper is presented as the protocol for Phase 1 and 2, why is phase 2 not clear as yet?	We apologize, but we do not understand what you mean we have tried to clarify phase II, page 5
#15 This paragraph could be summarised with less focus on patients. The reporting of the literature in this paragraph and ones below is very descriptive. it could be strengthened by inclusion of data to support statements	√. We have removed parts of the description of the patient group, page 6
# 16 It is not good practice to cite a string of refs without knowing what aspects of a statement relates to which ref. I would reduce the number of refs and use 1 or 2 key (most robust) ones to support the point being made	We agree and have clarified the references.
# 17 Won't this depend on the outcomes being targeted and so multiple approaches may be required?	We agree, we have rewritten the text on page 7
# 18 How can this sentence be supported when the mode of delivery and the framework for the intervention - i.e. Carer esupport has been decided?	We have clarified that ICs are involved in the development of both the content and the features of Carer eSupport. We develop the intervention, but it is in the RCT that we will test

	our hypothesis on whether the intervention has an effect or not, page 8
# 19 The aims stated earlier are to develop and test feasibility and acceptability. Suggest revising the document to be consistent throughout	We have clarified the aim with “The main aim of this study is to describe the development of an internet-based intervention (Carer eSupport) to improve ICs’ ability to support individuals with HNC and the testing of the Carer eSupport’s feasibility and acceptability. An additional aim is to explore HCPs’ perceptions of the content and design of Carer eSupport from a clinical perspective”, please see page 9
# 20 By articulating feasibility & acceptability at the end of the sentence, it suggests that phases 1 and 2 will also address improving IC health status, but this will be the future RCT? The paper would be strengthened by clarity and consistency of articulated aims and outcomes. What are the research questions driving phases 1 and 2? In health services user co-design- the views of HCPs and consumers are given equal status and brought together as part of the design process.	Please see the answer above. We have clarified in the introduction and in the methods that Phase I and II are preceding a forthcoming RCT. You raise an important point. However, we do not develop a service in health care, we develop an intervention that is to be evaluated in a RCT regarding the effects on outcomes for ICs. As we have clarified in the introduction, if the results of the upcoming RCT have an effect, we will try to implement the intervention in clinical activities.
#21 This suggests that HCP views are given primacy over consumers. The consumers develop and then the HCPs get to revise? Is this correct? If so, this is not user-centred design.	We have revised the text on pages 9 and 10.
#22 These questions seem leading? Should these be- did you face any major challenges; did you get any support etc	We agree that the questions are leading and could have been more open but the interviews had become less focused therefore we chose more targeted questions to catch the perception of the support ICs need page 10.
#23 How able will people be to answer this without having opportunity to see the intervention developed first?	We were interested in participants' general perception of AI and therefore we asked the question. This was done by providing samples of AI features.

	That part of the interview was conducted by experts in the field, page 10.
#24 As above- Do ICs of people with H&N cancers need support?	We have decided to conduct this study as the literature suggests that this group needs more support than the health care provides today. You have right even here, that the questions are leading and that the questions could have been more open.
# 25 Why use inductive & deductive- how will this be applied and to what data?	We agree that a detailed description of analyses is important. However, this is a study protocol describing the planning but not the final conduct of the studies, which may be slightly changed due to reasons that cannot be fully anticipated. A more detailed description of the analysis, and a discussion regarding the robustness, will be reported in future reports, please see page 11
# 26 These data will be critical to development of the intervention, it would be good to have more detail to assess robustness	
# 27 please provide more detail about this	11 A short description of the user-centered design methodology has been added to page 12.
# 28 This seems in antithesis of user-centred design- where patients/ICs voices should be equal. Why not have hcps and ICs working together in focus groups/workshops?	See the revised text on page 12.
# 29 As above- if user-centred design why are HCPs given priority?	12 This has been clarified on page 13.
# 30 The MRC framework sets out these components but why not use a feasibility framework- eg Proctor or Peters. Is the intent here to assess feasibility components for design of a future RCT or clinical utility feasibility components?	We have added clarification of why phases I and II will be conducted, in the introduction.
# 31 So its feasibility for RCT design? This needs to be clear right from the outset of the paper. The Cocks paper refers to pilot RCTs	12 This has been clarified in the introduction and the methods, page 12.
# 32 This is confusing as this seems to be more about clinical utility than feasibility design for an RCT	12 Please see above.

# 33 If this has been decided already- why undertake the development phase? Or is this the framework that the content developed and agreed in phase 1 will fit into?	12 We have added a clarification that the description of Carer eSupport is tentative, page 13.
# 34 Is phase 2 a mixed methods study? If so, what kind of design eg concurrent, sequential etc	Please see the revised text on page 12
# 35 These are feasibility elements to demonstrate feasibility of undertaking an RCT. Again, the intent of "feasibility" could be more clearly defined for the paper. How will the qualitative data be used to establish relevance etc?	A clarification has been made, on page 14.
# 36 Why not test relevance etc., before trying to establish recruitment/adherence/attrition etc. as these will have significant impact on pilot RCT recruitment feasibility outcomes?	We have clarified how ICs are involved in the development of the content and features of Carer eSupport, page 12. Thus, relevance, etc. is assessed by IC experts throughout the development of Carer eSupport.
# 37 I am not sure how relevant all the information below is in this paper, as it seems to be about establishing choice of measures for the RCT. Was that an aim of development and feasibility testing?	We think it is valuable information so we leave it to remain. We have added a clarification on page 14.
# 38 There is little information here to assess the rigour or appropriateness of the analysis plan	We have added a short description regarding how the rigorousness of the analysis may be confirmed. However, as stated above, this is a study protocol describing the planning but not the final conduction of the studies, which may be slightly changed due to reasons that cannot be fully anticipated, page 15.
# 39 If so, it would be good to explain why assess capacity for recruitment/suitability of questionnaires/adherence before establishing whether the content is right. Setting clear research questions at the outset will help the reader follow rationale of the components of what's being done.	We have obtained opinions of the content from ICs and HCP through FGDs. In addition, we have an ongoing discussion during the development of Carer eSupport with our expert group consisting of ICs. When we had performed the feasibility study, we will get further opinions of ICs, on the content and also on the questionnaires and Carer eSupport.

	As described and recommended in MRC guidelines, we will do an internal pilot test at the beginning of the forthcoming RCT. That will facilitate the understanding of the research.
# 40 Isn't this the purpose of ICs being recruited to the Focus Groups?	Involvement of the target population during the entire project is highly recommended. We have therefore formed an expert group of ICs to inform the research team during Phase I and Phase II and during the forthcoming RCT.
# 41 This statement suggests clinical utility testing rather than establishing feasibility of undertaking an RCT - defining feasibility at the outset of the paper will be very helpful	Please see the revised text, page 16